# H3K4me1 Modification Functions in Caste Differentiation in Honey Bees

**DOI:** 10.3390/ijms24076217

**Published:** 2023-03-25

**Authors:** Yong Zhang, Zhen Li, Xujiang He, Zilong Wang, Zhijiang Zeng

**Affiliations:** 1Honeybee Research Institute, Jiangxi Agricultural University, Nanchang 330045, China; yongzhang1994@sina.com (Y.Z.); zhenli1995@sina.com (Z.L.); hexujiang3@163.com (X.H.); wzlcqbb@126.com (Z.W.); 2Jiangxi Province Key Laboratory of Honeybee Biology and Beekeeping, Nanchang 330045, China

**Keywords:** histone modification, gene expression, methylation, caste-specific

## Abstract

Honey bees are important species for the study of epigenetics. Female honey bee larvae with the same genotype can develop into phenotypically distinct organisms (sterile workers and fertile queens) depending on conditions such as diet. Previous studies have shown that DNA methylation and histone modification can establish distinct gene expression patterns, leading to caste differentiation. It is unclear whether the histone methylation modification H3K4me1 can also impact caste differentiation. In this study, we analyzed genome-wide H3K4me1 modifications in both queen and worker larvae and found that H3K4me1 marks are more abundant in worker larvae than in queen larvae at both the second and fourth instars, and many genes associated with caste differentiation are differentially methylated. Notably, caste-specific H3K4me1 in promoter regions can direct worker development. Thus, our results suggest that H3K4me1 modification may act as an important regulatory factor in the establishment and maintenance of caste-specific transcriptional programs in honey bees; however, the potential influence of other epigenetic modifications cannot be excluded.

## 1. Introduction

Honey bees are eusocial insects and an important model organism for studies of caste development and caste differentiation in social insects. Their division of labor is mainly based on the differentiation of castes (queen and worker) [1]. After divergence, queens and workers have different morphological, physiological, behavioral, and longevity-related traits, despite sharing the same genome [2]. The mechanism underlying caste differentiation is not fully understood. However, there is evidence that differences in nutritional status between queens and workers modulate caste differentiation by altering DNA methylation patterns [3,4,5,6]. In addition, various signaling pathways, such as the Wnt signaling pathway [7], the target of rapamycin (TOR) nutrient sensing pathway [8,9], and the mitogen-activated protein kinase (MAPK) signaling pathway [10], are related to honey bee caste differentiation.

DNA and histone modifications are thought to primarily affect transcriptional events. The establishment, maintenance, and regulation of transcriptional programs during development depend on chromatin plasticity [11]. Recent evidence suggests that chromatin-based epigenetic mechanisms can influence nutrient-mediated caste differentiation in honey bees. RNAi knockdown of the DNA methyltransferase *DNMT3* has a jelly-like effect on developmental trajectories, resulting in a significantly higher proportion of queens with fully developed ovaries [12]. Differences in DNA methylation also influence the alternative development of queens and workers [13]. H3K27ac has been shown to be a key chromatin modification, and the caste-specific region of intronic H3K27ac directs the worker caste [11]. However, for H3K4me3 and H3K36me3, there is no evidence that modification in specific regions can direct caste development [11]. Histone methylation is methylation that occurs on the N-terminal lysine (K) [14] or arginine [15] residues of H3 and H4 histones. Like histone acetylation, histone methylation contributes to almost all biological processes, from DNA repair, the cell cycle, stress responses, and transcription to development, differentiation, and aging [16,17,18,19,20]. It can also regulate the lifespan of model organisms such as rats [21], *Caenorhabditis elegans* [22], and *Drosophila melanogaster* [23], and even have a transgenerational effect on lifespan [24,25]. However, there is no evidence that histone methylation contributes to honey bee development.

Hundreds of genes involved in caste differentiation have been identified in honey bees [13,26]. Differences in chromatin levels may lead to differences in gene expression. Histone acetylation influences caste differentiation in honey bees [11]. However, the role of histone methylation in honey bee caste differentiation has not been determined. We performed the first genome-wide analysis of the distribution of H3K4me1 before (2nd instar) and after (4th instar) critical time points (3rd instar) [1,27] in honey bee caste differentiation. Compared with the 2nd instar, the numbers of differentially expressed genes (DEGs) and H3K4me1 markers were significantly increased at the 4th instar, and gene expression was negatively correlated with H3K4me1. Further analysis revealed that the chromatin patterns of queens and workers differed significantly at the 4th instar. H3K4me1 modification promotes larval development towards worker bees. These findings illustrate the important role of H3K4me1 in honey bee larval development and caste differentiation.

## 2. Results

### 2.1. H3K4me1 Modifications in the Honey Bee Are Enriched in Transcribed Regions

To investigate the chromatin structure of honey bees, we determined, for the first time, the genome-wide distribution of H3K4me1. We detected H3K4me1 enrichment around the transcriptional start sites (TSS) of genes in both queen larvae and worker larvae (Appendix A). These results are consistent with those for other species, including mammals, invertebrates, and plants [28,29,30]. Over 8 G of data were generated for all samples, with a mapping value and Q30 value of greater than 90% (Appendix A). These results indicate that the sequencing results are reliable.

We further evaluated caste-specific patterns in the distribution of H3K4me1. In particular, we divided H3K4me1 into unique peaks (i.e., peaks detected in all three replicates in one sample, but not in any of the three replicates in the other sample) and differential peaks (*p* < 0.05 and fold change > 2). In 2Q (2nd instar queen) vs. 2W (2nd instar worker), we identified 36 unique H3K4me1 peaks in 2Q and 380 in 2W (Figure 1A). In 4Q vs. 4W, we identified 347 unique H3K4me1 peaks in 4Q and 1185 in 4W (Figure 1A). There were significantly more unique peaks for workers than queens at both the 2nd and 4th instars (Appendix A). There was no significant difference in the relative proportions of the positions of unique peaks between 2Q and 2W (Figure 1B; χ^2^ = 2.66, *p* = 0.27, chi-squared test). However, there was a difference in the distribution of unique peaks between 4Q and 4W (Figure 1C; χ^2^ = 80.18, *p* = 3.88 × 10^−18^, chi-squared test); 60% of the unique peaks in 4W were enriched in the promoter region, while 61% of the unique peaks in 4Q were enriched in intronic regions. The data for differential peaks between queens and workers were consistent with the results for unique peaks. In 2Q vs. 2W, we identified 18 differential H3K4me1 peaks in 2Q and 326 in 2W (Figure 1D). In 4Q (4th instar queen) vs. 4W (4th instar worker), we identified 432 differential H3K4me1 peaks in 4Q and 2233 in 4W (Figure 1D). There was no significant difference in the distribution of differential peaks between 2Q and 2W (Figure 1E; χ^2^ = 4.57, *p* = 0.10, chi-squared test). However, there was a difference in the distribution of differential peaks between 4Q and 4W (Figure 1F; χ^2^ = 45.32, *p* = 1.44 × 10^−10^, chi-squared test); 55% of unique peaks in workers were found in promoter regions, while 54% of unique peaks in queens were found in intronic regions. In order to study the functional significance of the caste-specific promoter H3K4me1, we performed a motif enrichment analysis using the MEME website (https://meme-suite.org/meme/tools/meme; accessed on 13 December 2022). The sequence of *Drosophila melanogaster* was used to annotate the honey bee motifs. We identified 2Q and 2W transcription factor binding sites that were significantly enriched, and the top three transcription factor binding sites were identical (Appendix A). However, the top five enriched transcription factor binding sites of 4Q and 4W were not the same (Appendix A).

### 2.2. Caste-Specific H3K4me1 Modification Patterns Correlate with Differential Gene Expression

After identifying caste-specific differences in the distribution of H3K4me1 modifications, we next evaluated whether the caste-specific distributions were associated with gene expression patterns according to caste. Principal component analysis of our RNA-seq data revealed strong separation between the two castes (Appendix A). In 2Q vs. 2W, we identified 666 DEGs and 296 differential peak-associated genes (DPGs), of which 28 genes were commonly differentially expressed in RNA-seq and ChIP-seq (Figure 2A). In 4Q vs. 4W, 2306 DEGs and 2004 DPGs were identified, of which 384 genes were commonly differentially expressed in RNA-seq and ChIP-seq (Figure 2A). In 2Q vs. 2W, there were 18 up-regulated peaks and 326 down-regulated peaks (Figure 2B), and 224 up-regulated genes and 452 down-regulated genes (Appendix A). In 2Q vs. 2W, there were 432 up-regulated peaks and 2233 down-regulated peaks (Figure 2B), and 1355 up-regulated genes and 951 down-regulated genes (Appendix A). We found that the genes enriched for the H3K4me1 modification showed significant expression differences based on transcriptional data (Figure 2C; 2Q vs. 2W, ρ = 0.07, *p* = 1.35 × 10^−27^, Spearman test; 4Q vs. 4W, ρ = 0.31, *p* = 3.93 × 10^−13^, Spearman test). In addition, there were significant correlations between differential H3K4me1 peak signals and transcript levels (Appendix A) and between DEGs and H3K4me1 peak signals (Appendix A). Based on the enrichment with enhancer-associated histone H3K4me1 modifications, transcription factor binding sites, promoter sites, and changes in gene expression, worker-specific H3K4me1-enriched regions are markers of active enhancers and play an important role in caste differentiation.

To investigate the effect of H3K4me1-modified genes on caste differentiation, a KEGG analysis was performed. We detected enrichment for distinct developmental processes in the two castes at both the 2nd and 4th instars. Among the unique peak-associated genes of 2W and 4W, eight KEGG pathways were related to honey bee caste differentiation, while among the unique peak-associated genes of 2Q and 4Q, there were only three and five KEGG pathways related to honey bee caste differentiation, respectively (Figure 3A–D). This was consistent with our previous analysis [31], again showing that H3K4me1 modification favors the development of honey bee larvae into worker bees. More DPGs in 4th instar larvae than in 2nd instar larvae were involved in caste differentiation (Figure 3E,F).

### 2.3. Caste Features of Worker Bees May Be Induced by H3K4me1

A representative gene involved in highly significant caste-specific changes in both H3K4me1 enrichment and gene expression analysis is shown in Figure 4A. *Juvenile hormone esterase* (*JHe*; LOC406066) is shown as an example of a physio-metabolic worker-specific gene [32], in which H3K4me1 enrichment differences are associated with the TSS. Similarly, the other two caste differentiation-related genes, *P450-6a17* and *IGF*, showed significant differences in gene expression and H3K4me1 enrichment in both TSS and gene ontology regions (Appendix A).

In addition, we selected five genes with established roles in the caste differentiation of honey bees for verification, namely *JHe* [32], *Vg* (vitellogenin, LOC406088) [33], *JHAMT* (juvenile hormone acid O-methyltransferase, LOC724216) [34], *Hex70a* (*hexamerin 70a*, LOC726848) [35], and *Hsp90* (*heat shock protein 90*, LOC408928) [36]. Our transcriptome results were consistent with those of previous studies [32,35,36,37], and trends in H3K4me1 enrichment were consistent with trends in transcriptome data, thus supporting the reliability of our results. Juvenile hormone (JH) is a master regulator of caste differentiation in honey bees [38]. Vitellogenin is an antagonist of JH. H3K4me1 enrichment in the *Vg* genes of workers was significantly higher than that in queens at the 2nd and 4th instars. These results further support the role of H3K4me1 modification in regulating transcription, and reveal the effect of H3K4me1 and transcript co-regulation on caste differentiation in honey bees.

## 3. Discussion

We used ChIP-seq to characterize genome-wide caste-specific chromatin patterns in honey bees and revealed the chromatin patterns related to reproductive division of labor in social insects. Combined with an RNA-seq analysis, the specific modification of H3K4me1 appeared after the critical time point for caste differentiation. There was no significant difference in H3K4me1 modification patterns between worker larvae and queen larvae until the irreversible stage of caste differentiation. A significant number of queen–worker chromatin differences are associated with caste-specific transcription. Importantly, a number of enhancers identified in caste-specific regions may be involved in honey bee caste differentiation.

Previous studies of *Drosophila melanogaster* [39,40], mice [41,42], and embryonic stem cells from humans and mice have demonstrated that changes in histone methylation affect development and regulate cell fate outcomes [43,44,45]. In honey bees, DNA methylation is predominantly present in gene body regions and the 5′ ends of genes [3,46,47]. Many previous studies have demonstrated the role of DNA methylation in caste differentiation. Different patterns of CpG methylation were detected between the queens and workers [3]. The results of the present study suggest that the developmental asymmetry between queen and worker larvae is associated with an asymmetry in H3K4me1 modification patterns. This is consistent with the distribution of DNA methylation in queen and worker bees [3]. Studies have shown that the degree of enrichment of H3K4me1 differs [48]. In addition, the differential histone methylation of transcription factors that regulate development is thought to alter cell fate decisions [44]. At the same time, the present study also showed that H3K4me1 modification can significantly regulate the transcript level of regulatory factors related to honey bee caste differentiation (such as *JHe*, *Vg*, *Hex70a*, and *JHAMT*). Signaling pathways such as FoxO and TOR are known to influence caste differentiation in honey bees [38,49], and the present study clearly shows that differential H3K4me1 levels also affect the transcription of genes in these signaling pathways. Notably, caste-specific H3K4me1-modified genes (such as *Hex70a* and *Vg*) are related to reproductive division of labor and nutrient metabolism [37,50,51]. These results suggest that H3K4me1 marks contribute to honey bee caste differentiation.

We found that H3K4me1 modification is closely involved in the modification of worker larvae, and is mainly involved in pathways related to honey bee caste differentiation. Caste-specific differences in H3K4me1 were mainly detected in the promoter and intronic regions. Queen-specific H3K4me1 was mainly located in intronic regions. In contrast, worker-specific H3K4me1 was mainly localized in promoter regions, close to transcription initiation sites. In addition, abundant caste-specific H3K4me1 in promoter regions was associated with high levels of caste-specific gene expression, suggesting that these regions play an important *cis*-regulatory role. Monomethylation of histone H3 at lysine 4 (H3K4me1) is a hallmark of activated enhancers in both vertebrates and invertebrates [28,52]. Enhancers are *cis*-regulatory DNA sequences and can increase the transcription of target genes. The opening of repressive enhancer–promoter loops leads to transcriptionally active enhancer–promoter regulation as a fundamental mechanism underlying differential transcriptional regulation [53,54]. The spatial organization of chromatin, including long-range enhancers adjacent to target promoters in *cis,* also modulates gene expression [55]. Therefore, we analyzed the caste-specific promoter region H3K4me1 for conserved transcription factor binding motifs. GATA2 and SPIB accounted for the most transcription factors in 4th instar queens and workers, respectively. Both GATA2 and SPIB can regulate development and cell differentiation [56,57,58,59]. Therefore, H3K4me1 modification may mediate caste-specific enhancer activation, thus directing larval development.

Taken together, we speculate that the worker-specific promoter H3K4me1 region has the hallmarks of an active enhancer. In addition, most worker genes enriched in the promoter H3K4me1 region are also transcription factors, suggesting that enhancers are associated with upstream and downstream genes during the development of worker castes. Queen-specific regions may also be caste-specific enhancers; however, further identification is needed.

## 4. Materials and Methods

### 4.1. Insects

Honey bees (*Apis mellifera*) were obtained from Jiangxi Agricultural University in 2020. Queens were restricted for 6 h (8 am to 2 pm) to a plastic frame designed by Pan et al. [60] to lay eggs in worker cells. The queen laid eggs on a removable plastic base, which was transferred to a plastic queen cell without touching the egg itself. Half of the eggs were transferred to queen cells at 2 pm on the second day after laying and before hatching, while the other half remained in the worker cells. All eggs (in both queen and worker cells) were cared for by workers. Eggs hatched on the third day after laying. To collect the queen and worker larvae at instars 2 and 4, larvae were sampled from both queen and worker cells in each of three colonies at 4 pm on days 5 and 7 after laying. Larvae were picked with sterilized tweezers and rinsed in ddH_2_O three times. Filter paper was used to drain the water from the larvae, and larvae were placed immediately in liquid nitrogen. Whole larvae were used for sequencing, as in previous studies, since they were very small (2nd instar worker and queen larvae: 2 mg/larva; 4th instar worker larvae: 12 mg/larva; 4th instar queen larvae: 25 mg/larva).

### 4.2. ChIP-seq Assay and Analysis

Chromatin immunoprecipitation was performed as described by Wojciechowski et al. [11], with slight modifications. Approximately 800 mg of larvae (320 larvae per sample in 2Q and 2W; 32 larvae per sample in 4Q and 4W) were cross-linked for 10 min in 1% ChIP-seq–grade formaldehyde. The H3K4me1 antibody (ab195391; Abcam, Cambridge, UK) was used for immunoprecipitation. The H3K4me1 library was sequenced (50 bp single-end reads or 150 bp paired-end reads) on an Illumina HiSeq3000 sequencer.

The genome assembly Amel_HAv3.1 (GCF_003254395.2) was downloaded from NCBI and indexed using Bowtie 2 (v2.3.3). ChIP-seq samples were mapped to this indexed genome using Bowtie 2 with the default parameters. Detailed mapping statistics for each sample are available in Appendix A.

### 4.3. RNA-seq Analysis

Hierarchical Indexing for Spliced Alignment of Transcripts (HISAT; version 2.1.5) was used to align the RNA-seq reads to the reference genome (Amel_HAv3.1). Expression levels were reported as FPKM values to normalize the length of the annotated transcripts and the total number of reads aligned to the transcriptome. Differential expression was analyzed using the DESeq2 R package (1.30.0). *p*-values were corrected for multiple comparisons with a false discovery rate (value < 0.05). Genes with *p*-adj ≤ 0.05 were defined as DEGs. Detailed statistics are available in Appendix A.

### 4.4. Verification of Gene Expression Differences by qRT-PCR

The 2nd and 4th instar larvae of queens and workers were sampled. Three samples were collected for each type of larvae, with each sample coming from a different colony. Therefore, a total of 12 larval samples across three different colonies were evaluated.

qRT-PCR was performed according to previously described methods [61]. Briefly, total RNAs were extracted using the TransZol Up Plus RNA Kit (TransGen Biotech, Beijing, China), and then transcribed into cDNA using a PrimeScript RT Reagent Kit (Takara, Kusatsu, Japan). *GAPDH* was used as a reference gene. The primer sequences were designed using Prime Primer 6.0 (Table 1). A 10 μL reaction system (5 μL of SYBR^®^Premix Ex Taq™ II, 3 μL of H_2_O, 1 μL of cDNA, 0.4 μL each of forward and reverse primers, and 0.2 μL of ROX) was established. The PCR conditions were as follows: 95 °C, 5 min; 94 °C, 2 min; 40 cycles (95 °C, 10 s, Tm, 15 s, 72 °C, 15 s); 72 °C, 10 min. To establish the melting curve of the qRT-PCR product, the primers were heated slowly with a gradual increase of 1 °C every 5 s from 72 °C to 99 °C. The data were analyzed using the 2^−ΔΔCT^ method.

## 5. Conclusions

Significant differences in H3K4me1 modification between queen and worker larvae were observed during caste differentiation in honey bees. Chromatin modification can regulate the transcription of the genes that determine the caste. Furthermore, H3K4me1 modification was closely involved in the regulation of the development of worker larvae, and may be an important modification in the worker development pathway. These findings clearly establish the contribution of histone methylation to honey bee development, and may contribute to further research on caste differentiation and developmental plasticity.

## Figures and Tables

**Figure 1 ijms-24-06217-f001:**
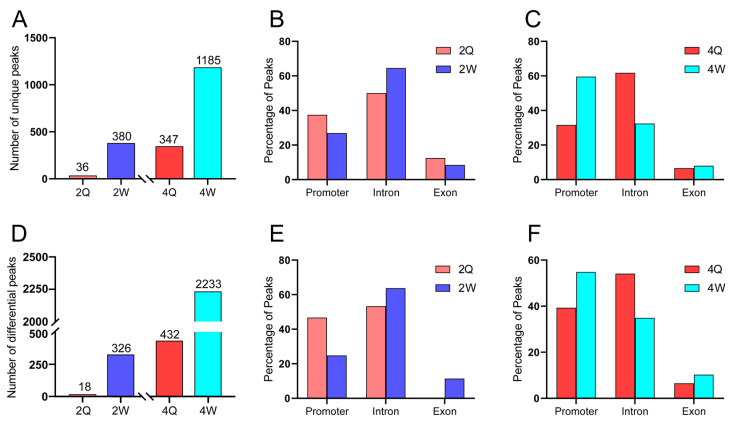
(**A**) Bar plot showing the number of unique H3K4me1 peaks in queen and worker larvae. (**B**,**C**) Bar plot showing the percentage of unique H3K4me1 ChIP-seq peaks within promoters, introns, and exons in 2Q vs. 2W and 4Q vs. 4W. (**D**) Bar plot showing the number of differential H3K4me1 peaks in queen and worker larvae. (**E**,**F**) Bar plot showing the percentage of differential H3K4me1 ChIP-seq peaks within promoters, introns, and exons in 2Q vs. 2W and 4Q vs. 4W.

**Figure 2 ijms-24-06217-f002:**
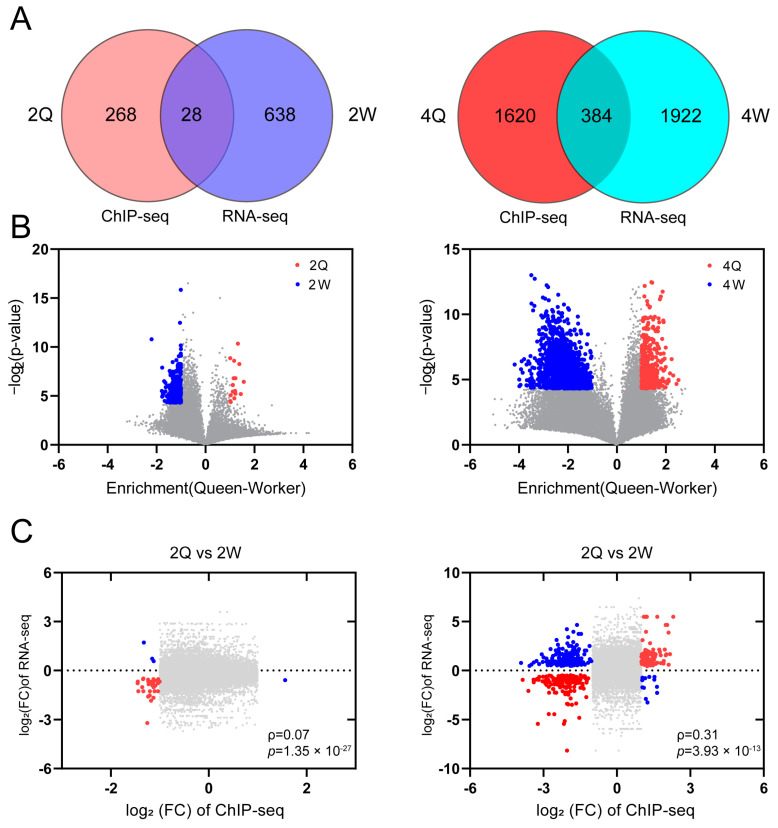
(**A**) Venn diagram showing overlap of differentially H3K4me1-modified genes and DEGs between queens and workers. (**B**) Volcano plot of the difference in enrichment between queens and workers against the negative log *p*-value for the H3K4me1 signal. Red areas indicate that the H3K4me1 modification is up-regulated in queens. Blue areas indicate that the H3K4me1 modification is up-regulated in workers. Gray indicates a lack of a significant difference (*p* > 0.05). (**C**) Scatter plots of the significant differences in expression, determined by ChIP-seq, between queen and worker larvae (*x*-axis) against the Log_2_FC of transcript expression between queen and worker larvae (*y*-axis). Red indicates that the differences determined by RNA-seq and ChIP-seq are in agreement. Blue indicates that the opposite patterns were obtained by RNA-seq and ChIP-seq. Gray indicates that there is no significant difference (*p* > 0.05).

**Figure 3 ijms-24-06217-f003:**
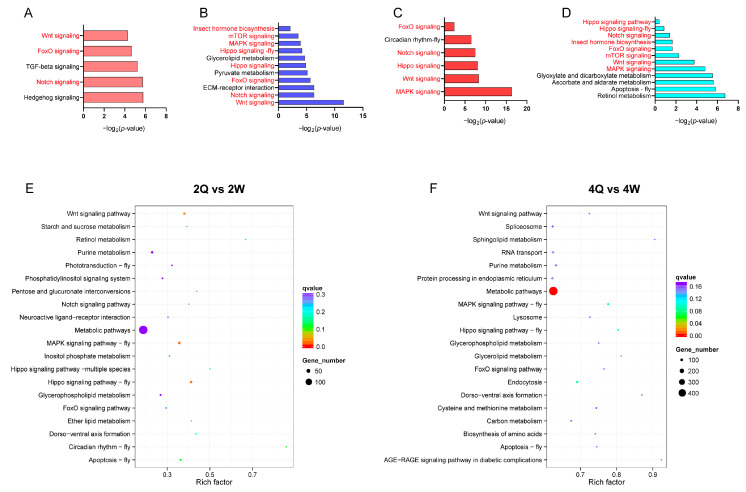
Negative log_2_ *p*-values for the honey bee caste differentiation-related KEGG pathways that are enriched with unique H3K4me1-related genes in 2Q (**A**), 2W (**B**), 4Q (**C**), and 4W (**D**). Pathways marked in red font are pathways associated with caste differentiation, while black font is not. (**E**) KEGG pathway enrichment analysis of differentially H3K4me1 peak-related genes in 2Q vs. 2W. (**F**) KEGG pathway enrichment of differentially H3K4me1 peak-related genes in 4Q vs. 4W.

**Figure 4 ijms-24-06217-f004:**
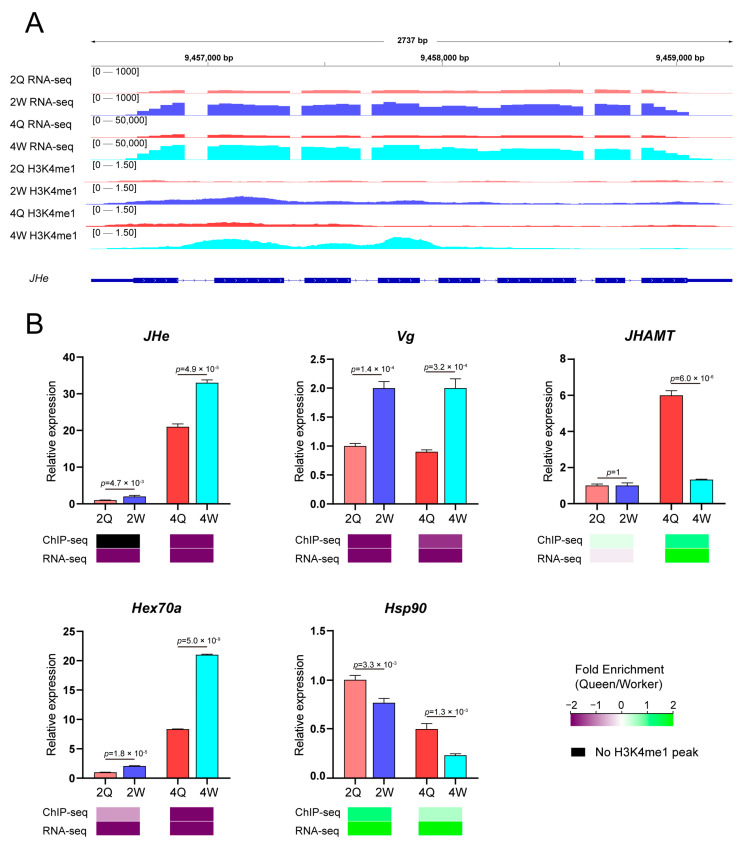
Function of H3K4me1 modification in caste differentiation by modulating the expression levels and modification abundance of genes. (**A**) Juvenile hormone esterase (*JHe*; LOC406066). The gene in this region reached genome-wide significance (*p* ≤ 0.01) at both the 2nd and 4th instars, and had a greater than two-fold difference in H3K4me1 modification. (**B**) RNA expression levels and H3K4me1 abundance of differentially expressed candidate caste-differentiation-related transcripts are shown. Expression levels are expressed as mean ± SEM relative to a reference gene (*GAPDH*; LOC726445) in three replications. Fold enrichment is the difference in abundance between queen larvae and worker larvae; green denotes a higher fold enrichment, while purple denotes a lower fold enrichment. Differences in relative expression levels were analyzed using *t*-tests.

**Table 1 ijms-24-06217-t001:** Primer sequences for quantitative qRT-PCR.

Genes	Forward Primer	Reverse Primer
*GAPDH*	GCTGGTTTCATCGATGGTTT	ACGATTTCGACCACCGTAAC
*JHe*	CTTTTCTCGCTTCCACAACC	TCCTGGTCCAGCAATGTGTA
*VG*	AAGACCAATCCACCGTTGAG	TGGTTCACGCTCCTAGCTTT
*JHAMT*	GGATTTGCCCAAAGACACAT	CGAGGATTCGCGTACAATTT
*Hex70a*	GAGGGTCAAGCATGGAACAT	GTTGTTCTTCGCCCAGAGAG
*Hsp90*	CTGAGAGTGACGCGAAGCTA	CTCCGGCATCTTTTCACAAT

## Data Availability

Raw sequencing reads for RNA-seq are available at SRA; accession: PRJNA770835. Raw sequencing reads for ChIP-seq are available at SRAl accession: PRJNA891375.

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
