# Peer review of "H3K4me1 Modification Functions in Caste Differentiation in Honey Bees"

_ijms, 2023, doi:10.3390/ijms24076217_

Round 1

Reviewer 1 Report

The study entitled, “H3K4me1 Modification Functions in Caste Differentiation in Honey Bee” encompasses original findings on the role of histone methylation modification (H3K4me1) in the caste differentiation of honey bee larvae that are genotypically similar but develops differentially into phenotypically two different castes as the sterile workers and fertile queens. The findings of the study are very interesting that indicates that modification of H3K4me1 play an important regulatory role in caste differentiation and maintenance of caste-specific transcriptional programs.

Overall, the article is well written. Rationale and objectives of the study are clearly stated in the introduction section. Sufficient details are provided on the protocol of the study. The results and discussion section are also of sufficient length reflecting a focused approach aimed at achieving the objectives of the study. I have the following recommendations:

The terminology, “2 instar, 4 instar, 3 instar etc.” used in the manuscript is not a standard format. It should be written as “2nd instar, 4th instar and 3rd instar etc.” OR the word “larva” or “larvae” may also be accompanied for more clarity such as, “2nd instar larva, 4th instar larva etc.”. Literally, the numbers, “2, 3, 4 etc.” written before the instar represent a count or number rather than a particular stage of larva that creates confusion. The terminology, “2Q, 2W or 4Q, 4W” used for Queen or Worker larvae may be maintained as such in the figures as well as in the text and there is no need of their modification.

The word, “Honey bees” and “honeybees” has been variously written in the manuscript. A uniform style should be followed either by writing it as, “honey bee” or “honeybee”.

Line 203, “Drosophila melanogaster” should be written in italics.

Conclusion should be modified and improved by stating the key findings of the study instead of mere suppositions.

Author Response

The study entitled, “H3K4me1 Modification Functions in Caste Differentiation in Honey Bee” encompasses original findings on the role of histone methylation modification (H3K4me1) in the caste differentiation of honey bee larvae that are genotypically similar but develops differentially into phenotypically two different castes as the sterile workers and fertile queens. The findings of the study are very interesting that indicates that modification of H3K4me1 play an important regulatory role in caste differentiation and maintenance of caste-specific transcriptional programs.

Overall, the article is well written. Rationale and objectives of the study are clearly stated in the introduction section. Sufficient details are provided on the protocol of the study. The results and discussion section are also of sufficient length reflecting a focused approach aimed at achieving the objectives of the study. I have the following recommendations:

Point 1: The terminology, “2 instar, 4 instar, 3 instar etc.” used in the manuscript is not a standard format. It should be written as “2nd instar, 4th instar and 3rd instar etc.” OR the word “larva” or “larvae” may also be accompanied for more clarity such as, “2nd instar larva, 4th instar larva etc.”. Literally, the numbers, “2, 3, 4 etc.” written before the instar represent a count or number rather than a particular stage of larva that creates confusion. The terminology, “2Q, 2W or 4Q, 4W” used for Queen or Worker larvae may be maintained as such in the figures as well as in the text and there is no need of their modification.

Response 1: Thanks for your suggestion, we have modified the content of the full text. "2nd instar, 4th instar, 3rd instar etc." was replaced by "2nd instar, 4th instar, 3rd instar etc." to avoid confusion. All the changes are marked in red.

Point 2: The word, “Honey bees” and “honeybees” has been variously written in the manuscript. A uniform style should be followed either by writing it as, “honey bee” or “honeybee”.

Response 2: Thanks for your suggestion, the words " Honey bees " and "honeybee" have been unified in the manuscript, all revised to "honey bee".

Point 3: Line 203, “Drosophila melanogaster” should be written in italics.

Response 3: Thanks for your suggestion, "Drosophila melanogaster "was written in italics.

Point 4: Conclusion should be modified and improved by stating the key findings of the study instead of mere suppositions.

Response 4: Thanks for your suggestion, we revised the conclusions. See line 300-307.

Reviewer 2 Report

The manuscript was very well prepared and contains relevant information for the academic community. However, authors should perform a careful grammar review. I suggest that a correction be made by a native speaker. Other suggestions are mentioned below:

1 - For your manuscript to have greater reach, use keywords different from those used in the title.

2 - The abstract perfectly summarized the content of the manuscript.

3 - Lines 57 and 58: "We infer that they may differ at the chromatin level leading to differences in gene expression." I suggest rewriting this sentence with a formal grammar.

4 - The results were brilliantly presented and the statistics were applied appropriately.

5 - The discussion was well explored, but authors should use formal grammar. Avoid the pronouns we and our.

6 - The results and discussion of the manuscript provide valuable data that should be explored in the conclusion. The conclusion needs to be rewritten and improved.

Author Response

Point 1: The manuscript was very well prepared and contains relevant information for the academic community. However, authors should perform a careful grammar review. I suggest that a correction be made by a native speaker. Other suggestions are mentioned below:

Response 1: Thanks for your comments. We asked MogoEdit to do language polishing for us.

Point 2: For your manuscript to have greater reach, use keywords different from those used in the title.

Response 2: Thanks for your comments. We changed keywords to "histone modification, gene expression, methylation, caste-specific".

Point 3: The abstract perfectly summarized the content of the manuscript.

Response 3: Thank you very much for your positive comments.

Point 4: Lines 57 and 58: "We infer that they may differ at the chromatin level leading to differences in gene expression." I suggest rewriting this sentence with a formal grammar.

Response 4: Thanks for your comments. We changed the sentence to a more formal grammar. See line 57.

Point 5: The results were brilliantly presented and the statistics were applied appropriately.

Response 5: Thank you very much for your positive comments.

Point 6: The discussion was well explored, but authors should use formal grammar. Avoid the pronouns we and our.

Response 6: Thank you very much for your positive comments. we modified our grammar to avoid "we" and "our" as much as possible.

Point 7: The results and discussion of the manuscript provide valuable data that should be explored in the conclusion. The conclusion needs to be rewritten and improved.

Response 7: Thanks for your comments, we revised the conclusions. See line 300-307